# The Association between Total Genotype Score and Athletic Performance in Weightlifters

**DOI:** 10.3390/genes13112091

**Published:** 2022-11-10

**Authors:** Hiroki Homma, Mika Saito, Aoto Saito, Ayumu Kozuma, Ryutaro Matsumoto, Shingo Matsumoto, Naoyuki Kobatake, Takanobu Okamoto, Koichi Nakazato, Tetsunari Nishiyama, Naoki Kikuchi

**Affiliations:** 1Faculty of Sport Science, Nippon Sport Science University, Tokyo 158-8508, Japan; 2Graduate School of Health and Sport Science, Nippon Sport Science University, Tokyo 158-8508, Japan; 3Faculty of Education, Ikuei University, Takasaki 370-0011, Japan; 4Faculty of Medical Science, Nippon Sport Science University, Tokyo 158-8508, Japan

**Keywords:** polymorphism, total genotype score, weightlifters, power athletes, genetics, gene markers, DNA, athletic status

## Abstract

This study aimed to investigate the relationship between power-oriented genetic polymorphisms and weightlifting status, create a total genotype score (TGS), and validate the association between TGS models and power-oriented athletes. First, 192 weightlifters and 416 controls were studied, and 12 polymorphisms that have previously been associated with strength, power status, and phenotype were genotyped using the TaqMan SNP genotyping assay. We calculated the TGS for the 12 polymorphisms using a PWM (power-oriented whole model) and for 6 of them using a WRM (weightlifting-related model) based on a case–control study. Second, the TGS of the WRM was compared for 177 strength and power athletes and 416 controls. There was no significant difference in the PWM score between weightlifters and the controls. Weightlifters and elite weightlifters had higher WRM scores than the controls. However, the WRM score had no association with weightlifting performance. There was no significant difference in the WRM between power-oriented athletes and the controls. Our study was able to create a TGS model for weightlifters based on case–control results. However, the TGS model could not be applied to other power-oriented athletes.

## 1. Introduction

Muscle strength and power are dependent on environmental and genetic factors and are required by athletes for successful competition. The genetic factor was estimated to have 66% athletic status heritability. Ahmetov et al. [1] reviewed the association between strength and power, athletic status, and gene markers, and reported at least 38 and 24 strength- and power-related markers. While a single polymorphism is partially responsible for muscle strength variation, the phenotypic influence of these single gene polymorphisms may be limited. Many studies have reported an association between sports performance and the *ACTN3* R577X polymorphism; however, this polymorphism may have a low contribution to athletic status and performance. Kikuchi et al. [2] showed that this polymorphism is responsible for 4.6% of the variability in relative peak power using the Wingate anaerobic test, while other studies have reported its contribution to be approximately 2% for muscle strength [3]. Therefore, genetic profiles that are more associated with athletic status and performance are needed.

Some studies have attempted to use the total genotype score (TGS) for the accurate prediction of athletic status and performance [4,5,6,7]. Ben-Zaken et al. created two polygenetic scores for endurance (endurance genetic distance score: EGDS, two and five polymorphisms) and power (power genetic distance score: PGDS, two and five polymorphisms) in related athletes and compared them with those of endurance and power athletes [5]. The PGDS2 and PGDS5 scores of power–speed athletes are significantly higher than their EGDS2 and EGDS5 scores and the PGDS2 and PGDS5 scores of controls. Similarly, the EGDS2 and EGDS5 scores of endurance athletes are significantly higher than their PGDS2 and PGDS5 scores and the EGDS2 and EGDS5 scores of controls. Therefore, polygenetic scores created for each athlete may indicate the athletes’ characteristics.

Several studies involving Japanese participants have not been able to replicate these results [8,9]. Miyamoto-Mikami et al. [8] calculated the TGS based on 21 polymorphisms which have been associated with sprint or power performance and their related phenotypes in 211 Japanese sprint and power athletes and 649 controls; however, the TGS was not able to predict the Japanese sprint and power athletes’ characteristics. In this study, genetic polymorphisms that report the association of sprint and power athlete status, sprint performance, and related phenotypes are used. However, some of these polymorphisms are not directly related to athletic performance factors such as lean body mass and free-fat mass [10,11]. In addition, previous studies recruited athletes in various athletic events involving power/sprint and endurance. To solve these problems, it is necessary to identify relevant genetic polymorphisms in a single athletic event that can clearly identify athletic performance. Weightlifting comprises snatch, clean, and jerk actions that are performed in under 1.0 s, reflecting an athlete’s strength and power [12,13]. Creating a more predictive TGS for strength/power athletes might be possible by using genetic polymorphisms associated with athletic performance and/or athletic status in weightlifters. 

Therefore, the purposes of this study are to (1) investigate the relationship between power-oriented genetic polymorphisms and weightlifting status; (2) create a TGS score using polymorphism; (3) examine the association between the TGS and performance in study 1; and (4) validate the applicability of association of the TGS with weightlifters onto power-oriented athletes in study 2.

## 2. Materials and Methods

### 2.1. Participants 

Overall, 192 Japanese weightlifters (113 men and 79 women) and 416 controls (151 men: height: 169.0 ± 6.3 cm, weight: 68.5 ± 9.6 kg, age: 51.0 ± 19.07 years; 265 women: height: 156.9 ± 8.1 cm, weight: 54.8 ± 8.1 kg, age: 53.0 ± 17.4 years) living in Tokyo and the surrounding areas were recruited in study 1. The weightlifters were divided into two groups: 67 international-level participants, who took part in international competitions such as world championships and Olympic games, and 125 national-level participants, who took part in the Japanese national competitions. In study 2, 177 strength and power athletes and 416 controls participated. The strength and power athlete group comprised 101 wrestlers (87 men, 14 women; 55 at international and 46 at national levels), 40 male powerlifters (16 at international and 39 at national levels), and 36 national-level throwers (17 men, 19 women).

### 2.2. Genotyping

Total DNA was extracted from saliva samples using an Oragene DNA Kit (DNA Genotek, ON, Canada) according to the manufacturer’s instructions. *ACE* I/D rs4341, *ACTN3* R577X rs1815739, *AGTR2* rs11091046, *ALDH2* rs671, *CHRNB3* rs4950, *CKM* rs8111989, *CNTFR* rs41274853, *FTO* rs9939609, *GALNTL6* rs558129, *IGF2* rs680, *MCT1* T1470A rs1049434, *NOS3* rs207044, *PPARGC1A* rs8192678, and *TRHR* rs7832552 polymorphisms were genotyped using a TaqMan SNP genotyping assay and a Real-Time PCR system (Applied Biosystems, Foster City, CA, USA). The Assay IDs for each polymorphism are provided in Appendix A.

### 2.3. Genotype Score

Genetic polymorphisms that were associated with power- or strength-oriented athletes were identified by searching for published original articles in PubMed and Google scholar (keywords: “Athlete”, “Power”, “Strength”, and “Status”). Polymorphisms with minor alleles have a frequency of more than 5% in the Japanese population according to jMorp (https://jmorp.megabank.tohoku.ac.jp/, accessed on 5 May 12021), because the sample size was limited. A given frequency of minor alleles was used for this study. Overall, 14 polymorphisms were found to be candidate genes associated with muscle strength and power. However, two polymorphisms were excluded in this study because the *AGTR2* rs11091046 polymorphism was located on the sex chromosome (Xq22-q23), the allele frequency was different in the sexes, and the *NOS3* rs2040744 polymorphism deviated from the Hardy–Weinberg equilibrium in the control and all participants and was different from the common frequency despite multiple analyses being performed. Therefore, we selected 12 polymorphisms. The TGS was classified into two models. First, a power-oriented model (PWM) comprising 12 candidate genes related to muscle strength and power, and the classification was based on Williams and Folland [14] and methods described by Miyamoto-Mikami et al. [8]. Specifically, we scored 0, 1, and 2 as “less optimal”, “intermediate”, and “optimal” genotypes, respectively. Second, a weightlifting-related model (WRM) was developed based on case–control results for each genetic polymorphism in weightlifters and controls. The genotypes that differed significantly for the genotype, dominant, or recessive models in all the weightlifters or the international group were in the WRM (Table 1). In the order of relevance of homotypes, we scored 0, 1, and 2 as “less optimal”, “intermediate”, and “optimal” genotypes with WRM. A score table of the genetic polymorphisms for each model is shown in Table 1. The genotype score was then totaled and converted to a scale of 0–100 for ease of interpretation. The formula for calculating TGS is as follows: $$\mathrm{TGS}=\frac{100}{24}\times \;(\mathrm{ACE}+\mathrm{ACTN}3+\mathrm{ALDH}2+\mathrm{CHRNB}3+\mathrm{CKM}+\mathrm{CNTFR}+\mathrm{FTO}+\mathrm{GALNTL}6+\mathrm{IGF}2+\mathrm{MCT}1\;+\mathrm{PPARGC}1A+\mathrm{TRHR})$$

### 2.4. Statistical Analyses

All statistical analyses were performed using the statistical software SPSS Statistics version 25 (IBM Japan, Tokyo, Japan) and SNPstats software (https://www.snpstats.net/, accessed on 1 July 2021). χ^2^ analysis was used to assess the Hardy–Weinberg equilibrium. Associations between the athlete status and each polymorphism were assessed using χ^2^ analysis under three genetic models (that is, a 3-genotype comparison: dominant, recessive, and additive). In addition, the comparisons of weightlifting and powerlifting performance between the genotypes were analyzed using a one-way analysis of variance (ANOVA). We compared the TGSs between weightlifters and controls using an unpaired *t*-test. Additionally, the comparison of the TGS between the three groups (that is, the control, and national and international levels) was carried out using ANOVA. Linear regression analysis was performed to estimate the degree of variance in the performance of weightlifters associated with the TGS. *p*-values < 0.05 were considered statistically significant.

## 3. Results

The genotype frequencies of all polymorphisms, excluding the *ACE* I/D, *CKM* rs8111989, and *CNTFR* rs41274853 polymorphisms in weightlifters, and the *CHRNB3* rs4950 and *TRHR* rs7832552 polymorphisms in the controls, were consistent with the Hardy–Weinberg equilibrium. The athletes were not often at equilibrium; hence, three polymorphisms were adopted in this study. Meanwhile, the frequencies of *CHRNB3* rs4950 and *TRHR* rs7832552 polymorphisms were equal according to jMorp; therefore, they were adopted in this study. The frequencies of each of the 12 polymorphisms in weightlifters are presented in Table 2 (see Appendix A for details). Only six polymorphisms (*ACE* I/D rs4341, *ACTN3* R577X rs1815739, *CHRNB3* rs4950, *CNTFR* rs41274853, *MCT1* T1470A rs1049434, and *PPARGC1A* rs8192678) were found to be associated with weightlifting athlete status.

The TGS based on the 12 polymorphisms in each model is shown in Figure 1A,B. All the PWM groups showed similar scores (controls: 43.6 ± 8.92, national: 42.78 ± 8.53, and international: 45.73 ± 9.55). Of the six genotypes associated with weightlifting status using the case–control study, the TGSs weightlifters were higher than those of the controls (controls: 32.93 ± 13.34, national: 35.53 ± 14.40, and international: 38.60 ± 15.16) and demonstrated a linear trend, as shown in Figure 2A,B. Weightlifters’ performance results based on the Wilks point (total records) were statistically significant for *ACTN3* R577X (RR + RX: 193.32 ± 27.25 vs. XX: 184.55 ± 27.61, *p* = 0.04) and *CNTFR* rs41274853 (CC: 185.32 ± 23.90 vs. TT + CT: 197 ± 30.30, *p* = 0.004). A trend (*p* < 0.1) was observed in the *ACE* I/D and *PPARGC1A* polymorphisms (Appendix A). However, regression analysis showed that the WRM was not a significant factor in performance (*p* = 0.167).

There was no significant difference in the case–control study between the six genotypes in power-oriented athletes (Appendix A). Furthermore, the TGS of the WRM was not significantly different between power-oriented athletes and the controls (controls: 32.3 ± 13.4, power-oriented athletes: 32.9 ± 13.3), as shown in Figure 3.

## 4. Discussion

In this study, we investigated the relationship between power-oriented genetic polymorphisms, weightlifting status, and performance; created the TGS score using polymorphism; and examined the associated between TGS and performance in study 1. Additionally, we validated the association of TGS with strength and power in athletes in study 2. The 12 polymorphisms were reported to have strength and power phenotypes in previous studies. Six of the twelve genetic polymorphisms were found to be associated with weightlifter status. There was no significant difference in the score of the 12 genetic polymorphisms between the weightlifters and controls. The WRM score of the six genetic polymorphisms associated with weightlifter status was found to be significantly higher in weightlifters than in the controls. However, the WRM exhibited no significant difference between score and performance.

Six of the twelve genetic polymorphisms were associated with weightlifter status; additionally, its polymorphisms, excluding *CHRNB3* rs4950, were reportedly associated with the muscle strength or power phenotype [15,16,17]. In particular, *ACE* I/D, *ACTN3* R577X, *CNTFR* rs41274853, and *PPARGC1A* rs8192678 in the WRM associated with elite weightlifters were associated with weightlifter performance. Furthermore, these genetic polymorphisms have also been reported to be associated with muscle strength and power in previous studies. *ACE* I/D polymorphisms are associated with ACE activation in the blood [18,19]. The II genotype was reported to have low ACE activity in muscle or blood with a higher percentage of type I muscle fibers [20]. On the other hand, several studies reported that the association of *ACE* I/D polymorphisms with muscle strength, power, and sprint athlete status was shown with the D allele. Wang et al. [21] examined gene frequencies in Caucasian and East Asian competitive swimmers. These results showed a higher frequency of the D allele in Caucasian athletes in short-distance events < 400 m and a higher frequency of the I allele in East Asian athletes, suggesting that the effects of genetic polymorphisms may differ between races. However, the direction of the associated allele from these studies is unknown, but appears to be related to athlete status. In this study, the direction of the relevant allele of the ACE gene differed between case–control and performance, although the reason is not clear. These findings suggest that caution should be exercised when using ACE to examine athletes’ status and performance.

ACTN3 was expressed in type II myofibers only [22], which is important for anchoring actin, and plays a regulatory role in coordinating muscle fiber contraction [23]. However, the XX genotype of *ACTN3* R577X did not express ACTN3 protein, and may have been affected by the phenotype. Many studies examining the association between the *ACTN3* R577X polymorphism and athletes have been performed. Tharabenjasin et al. [24] conducted a meta-analysis of 38 meta-analyses of elite power-oriented athletes published in 2018, and reported that RR and RX genotypes are associated with elite power athlete status. The *CNTFR* rs41274853 polymorphism is related to the sprint or power performance and muscle phenotypes [25,26,27]. The reason could be that this polymorphism is located in the 3′-UTR of CNTFR, and it may affect the binding of micro-RNA [27]. Miyamoto-Mikami et al. [27] reported that leg extension power and vertical jump height were higher in the TT genotype of *CNTFR* rs41274853 polymorphism than in the CC + CT genotype in young or middle-aged males. Muscle strength is directly related to weightlifting performance [13]; therefore, these polymorphisms may be associated with weightlifting status or performance. A previous study reported that mRNA of PPARGC1A expression was lower in those with the A-allele of the *PPARGC1A* polymorphism than in those with the GG genotype [28]. Additionally, PGC-1 knock-out mice showed a myofiber transition from type I and IIa fibers to type IIx and IIb [29]. Gineviciene et al. [30] examined the gene frequency of the *PPARGC1A* rs8192678 polymorphism in 114 Russian and 47 Lithuanian strength- and power-oriented athletes and found that the frequency of the AA genotype in strength- and power-oriented athletes was higher than that in the controls.

In this study, a TGS was created using these genetic polymorphisms. However, no significant differences were found between the controls and weightlifters in the WPM, and this may be because 12 polymorphisms were used. Each increase in polymorphisms causes a decrease in the probability of a perfect TGS and may include polymorphisms that do not reflect weightlifting performance. Earlier studies that related the TGS with power-oriented athletes’ status created a TGS with three to seven genetic polymorphisms [5,6,31]. Hence, the WRM was created from six genetic polymorphisms, which may have shown significant differences.

Guilherme and Lancha [7] compared TGSs created via several combinations of 10 genetic polymorphisms in 368 athletes and 818 controls. According to the study, scores above 60 were more common in elite athletes than in non-athlete controls [7]. Consequently, athletes with higher scores may have had more genetic traits than the controls. In this study, the frequency distribution of the participants according to the WRM showed that weightlifters are more overexposed than the controls, with scores ≥ 50.0 (WRM: controls: 15.62%; international: 28.36%; *p* = 0.02). Based on our results and those of previous studies, it may be necessary to have more than a given score (e.g., 50<) to be successful as an elite athlete.

In this study, the TGS model created for weightlifters could not be reproduced in other power-oriented athletes (wrestlers, powerlifters, and throwers). One of the reasons is the differences in the characteristics and energy supply systems, which could not reflect the same genetic characteristics. Wrestling is a competition of two rounds of 3 min with a 30 sec rest between rounds; this makes it a high-intensity, intermittent exercise requiring physical fitness, such as maximal muscle strength and endurance, as well as anaerobic and aerobic capacity [32]. In powerlifting, all the weightlifting is completed in three events—squat, deadlift, and bench press—and performance improvement may be associated with maximal muscle strength, not fast speed. These are athletes with the same strength and power but with different abilities required for weightlifting competitions, and they may have different genetic characteristics; hence, the TGS model created for weightlifters may not have been applicable to other power-oriented athletes. A previous study also reported different genetic characteristics in a similar event and indicated that different events should not be combined in sports genetic studies, or should at least be examined carefully [33].

A limitation of this study is that it was necessary to consider multiple comparisons and conduct Bonferroni adjustments. The only genetic polymorphism that remained significant after correction for multiple comparisons was the *CNTFR* rs41274853 polymorphism (*p* = 0.007). In addition, there may also be different directions of association of polymorphisms in case–control and performance, as in the ACE I/D polymorphism. However, it is necessary to examine the reproducibility of this study in other weightlifter cohorts to confirm its reliability. The polymorphisms examined in the present study represent only a small percentage of the polymorphisms possessed by the human genome. Therefore, it is necessary to perform a comprehensive analysis of polymorphisms using a genome-wide association study which can analyze the entire human genome, and enrichment analysis which is able to analyze their function. In addition, the composite effect of hundreds to thousands of genetic polymorphisms using the polygenic risk score should be examined to create a TGS that is more related to muscle strength and power.

## 5. Conclusions

Six genetic polymorphisms related to weightlifter status could be identified. The TGS model generated from those genetic polymorphisms was higher in elite weightlifters compared to the controls. However, the TGS model using six polymorphisms could not be applied to weightlifting performance and status in other power athletes.

## Figures and Tables

**Figure 1 genes-13-02091-f001:**
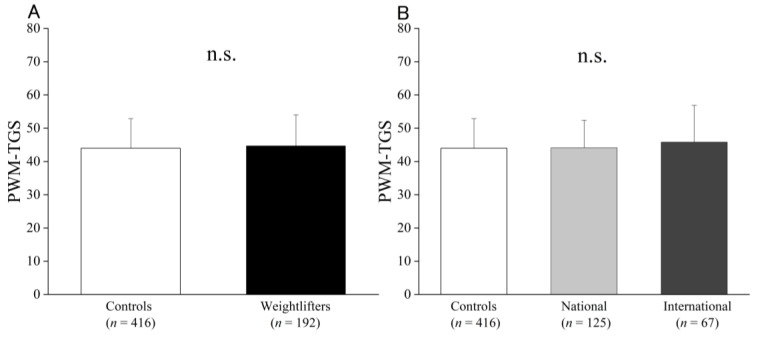
PWM of the controls and weightlifters (**A**) and of the controls and the national and international groups (**B**). The values are presented as mean ± SD.

**Figure 2 genes-13-02091-f002:**
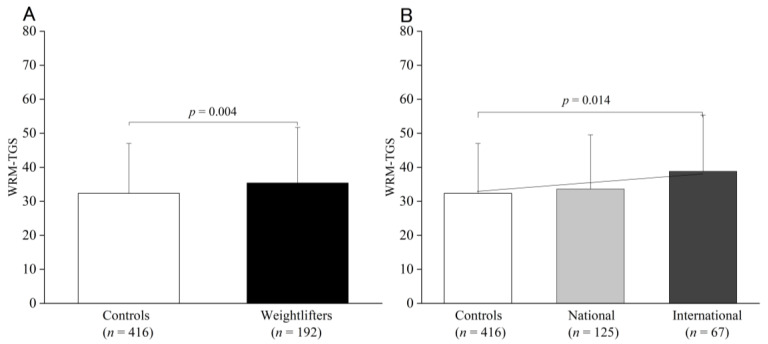
WRM of controls and weightlifters (**A**) and of the controls and the national and international groups (**B**). The values are presented as mean ± SD.

**Figure 3 genes-13-02091-f003:**
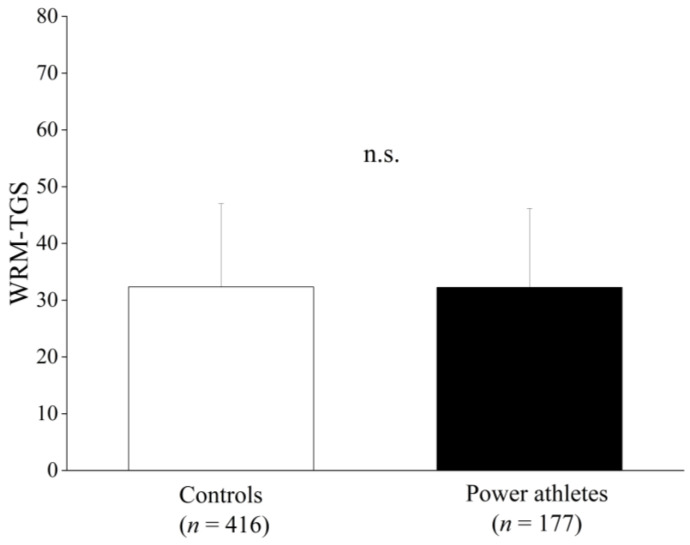
WRM of power-oriented athletes and the controls. The values are presented as mean ± SD.

**Table 1 genes-13-02091-t001:** Polymorphisms of each model in weightlifters.

Gene Symbol	rs Number	Polymorphism	Genotype Score Model					
			PWM			WRM		
*ACE*	rs4341	I/D	DD = 0	ID = 1	II = 2	DD = 2	ID = 1	II = 0
*ACTN3*	rs1815739	C > T	CC = 2	TC = 1	TT = 0	CC = 2	TC = 1	TT = 0
*ALDH2*	rs671	G > A	GG = 2	GA = 1	AA = 0			
*CHRNB3*	rs4950	A > G	AA = 2	GA = 1	GG = 0	AA = 0	GA = 1	GG = 2
*CKM*	rs8111989	A > G	AA = 0	GA = 1	GG = 2			
*CNTFR*	rs41274853	C > T	CC = 0	CT = 1	TT = 2	CC = 0	CT = 1	TT = 2
*FTO*	rs9939609	T > A	TT = 0	TA = 1	AA = 2			
*GALNT6*	rs558129	C > T	CC = 0	CT = 1	TT = 2			
*IGF2*	rs680	G > A	GG = 2	GA = 1	AA = 0			
*MCT1*	rs1049434	A > T	AA = 0	TA = 1	TT =2	AA = 0	TA = 1	TT = 2
*PPARGC1A*	rs8192678	A > G	AA = 2	GA = 1	GG = 0	AA = 2	TA = 1	GG = 0
*TRHR*	rs7832552	T > C	TT = 2	CT = 1	CC = 0			

PWM: power-oriented whole model; WRM: weightlifting-related model.

**Table 2 genes-13-02091-t002:** Frequency of 12 polymorphisms in weightlifters and controls.

Gene	Genotype	n(%)						*p*-Value					
		All Weightlifters		International		Controls		All Weightlifters vs. Controls			International vs. Controls		
		(*n* = 192)		(*n* = 67)		(*n* = 416)		Dominant	Recessive	Allele	Dominant	Recessive	Allele
*ACE*	DD	40	(21)	12	(18)	49	(12)	DD/II + ID	DD + ID/II	D/I	DD/II + ID	DD + ID/II	D/I
rs4341	ID	79	(41)	22	(33)	200	(48)	**0.003**	0.062	0.060	0.161	0.160	0.738
	II	73	(38)	33	(49)	167	(40)						
*ACTN3*	RR	49	(25)	21	(32)	84	(20)	RR + RX/XX	RR/RX + XX	R/X	RR + RX/XX	RR/RX + XX	R/X
rs1815739	RX	86	(45)	31	(46)	226	(55)	0.276	0.139	0.850	0.580	**0.040**	0.120
	XX	57	(30)	15	(22)	106	(25)						
*ALDH2*	GG	99	(51)	34	(51)	232	(56)	GG + GA/AA	GG/GA + AA	G/A	GG + GA/AA	GG/GA + AA	G/A
rs671	GA	76	(40)	26	(39)	149	(36)	0.332	0.396	0.390	0.443	0.390	0.390
	AA	17	(9)	7	(10)	35	(8)						
*CHRNB3*	AA	127	(66)	47	(70)	323	(77)	AA/GA + GG)	AA + GA/GG	A/G	AA/GA + GG)	AA + GA/GG	A/G
rs4950	GA	61	(32)	19	(29)	77	(19)	**0.002**	0.257	0.020	0.170	0.331	0.418
	GG	4	(2)	1	(1)	16	(4)						
*CKM*	AA	152	(79)	52	(78)	308	(74)	AA/GA + GG)	AA + GA/GG	A/G	AA/GA + GG)	AA + GA/GG	A/G
rs8111989	GA	34	(18)	13	(19)	98	(24)	0.600	0.170	0.290	0.776	0.533	0.640
	GG	6	(3)	2	(3)	10	(2)						
*CNTFR*	CC	104	(54)	26	(39)	210	(50)	CC/CT + TT	CC + CT/TT	C/T	CC/CT + TT	CC + CT/TT	C/T
rs41274853	CT	64	(33)	28	(42)	170	(41)	0.397	0.139	0.970	0.076	**0.007**	**0.008**
	TT	24	(13)	13	(19)	36	(9)						
*FTO*	TT	118	(61)	44	(66)	269	(65)	TT/TA + AA	TT + TA/AA	T/A	TT/TA + AA	TT + TA/AA	T/A
rs9989609	TA	65	(34)	21	(31)	130	(31)	0.445	0.733	0.440	0.770	0.623	0.936
	AA	9	(5)	2	(3)	17	(4)						
*GALNTL6*	CC	137	(71)	48	(72)	304	(73)	CC/CT + TT	CC + CT/TT	C/T	CC/CT + TT	CC + CT/TT	C/T
rs558129	CT	49	(26)	17	(25)	101	(24)	0.658	0.738	0.610	0.806	0.872	0.780
	TT	6	(3)	2	(3)	11	(3)						
*IGF2* r	CC	80	(42)	25	(37)	143	(34)	CC/CT + TT	CC + CT/TT	C/T	CC/CT + TT	CC + CT/TT	C/T
s680	CT	83	(43)	32	(48)	193	(46)	0.217	0.217	0.050	0.400	0.400	0.430
	TT	29	(15)	10	(15)	80	(20)						
*MCT1*	AA	75	(39)	28	(42)	199	(48)	AA/TA + TT	AA/TA vs. TT	A/T	AA/TA + TT	AA/TA vs. TT	A/T
rs1049434	TA	91	(47)	30	(45)	175	(42)	**0.043**	0.210	0.035	0.357	0.409	0.270
	TT	26	(14)	9	(13)	42	(10)						
*PPARGC1A*	AA	35	(18)	13	(19)	79	(19)	AA/GA + GG)	AA + GA/GG	A/G	AA/GA + GG)	AA + GA/GG	A/G
rs8192678	GA	102	(53)	38	(57)	184	(44)	0.823	**0.049**	0.220	0.936	**0.039**	0.147
	GG	55	(29)	16	(24)	153	(37)						
*TRHR*	TT	52	(27)	13	(19)	129	(31)	TT/CT + CC	TT + CT/CC	T/C	TT/CT + CC	TT + CT/CC	T/C
rs7832552	CT	98	(51)	32	(48)	175	(42)	0.325	0.183	0.850	0.053	0.310	0.059
	CC	42	(22)	22	(33)	112	(27)						

Full names of the genes are: ACE: angiotensin I-converting enzyme; ACTN3: α-actinin-3; ALDH2: mitochondrial aldehyde dehydrogenase 2; CHRNB3: cholinergic receptor, nicotinic, β 3; CKM: Muscle-Specific Creatine Kinase; CNTFR: cilliary neurotrophic factor receptor; FTO: fat mass and obesity-associated gene; GALNTL6: N-acetylgalactosaminyltransferase-like 6 gene; IGF2: insulin-like growth factor2; MCT1: monocarboxylate transporter 1 gene; PPARGC1A: peroxisome proliferator-activated receptor γ coactivator 1 α; and TRHR: thyrotropin-releasing hormone receptor. The ACE I/D polymorphism (rs4340) was determined using the rs4341 polymorphism, which is in perfect LD with the ACE I/D polymorphism (rs4340). PWM: power-oriented whole model; WRM: weightlifting-related model.

## Data Availability

The data presented in this study are available on request from the corresponding author.

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
