# Peer review of "The Association between Total Genotype Score and Athletic Performance in Weightlifters"

_genes, 2022, doi:10.3390/genes13112091_

Round 1
Reviewer 1 Report
The aim of this study was to explore the genetic score in Japanese athletes, particularly weightlifters. The proposal is interesting and well designed, but there are a number of points that need to be adjusted (listed below).
Abstract
Lines 14-15: It appears that the second and third proposals are the same (study/validate the derived models, WCM and WPM, in an independent cohort of strength/power athletes). If this is true, the two can be combined to facilitate interpretation.
Introduction
Lines 32-34 (While a single polymorphism is responsible for muscle strength variation…): It is important to note that a single polymorphism is partially responsible for muscle strength variation.
Materials and Methods
Lines 71-72: How many men and women are there in the control group? (similar to that described for athletes)
Line 75: What is the competitive level of the athletes included in study 2? (international level or only national). It is worth noting that a general description of the study design may be interesting. For example, highlighting that the athletes from study 2 were used only to validate the score derived from study 1.
Lines 81-83: Were only polymorphisms with minor allele frequency > 5% in the Japanese population considered? It wasn't clear.
Lines 83-84 or 88 (Overall, 14 polymorphisms were found to be candidate genes associated with muscle strength and power or Therefore, we selected 12 polymorphisms): Here, it would be interesting to write the name of the genes + rs# of each variant used.
Results
Lines 143-144 (…only three polymorphisms were adopted in this study): It is confusing to understand why it is saying that only 3 polymorphisms were adopted in the study, while the TGS was calculated with the 12 polymorphisms. What was actually considered or disregarded in the main analyzes based on the Hardy-Weinberg equilibrium could be better described.
Figures: The axes title was difficult to read.
Line 154 (…no significant difference between weightlifters and controls (control: 39.18 ± 19.33, weightlifter: 40.62 ± 20.8) as shown in Fig. 1 E and F): The description is wrong because in Fig 1 F there are data with statistical difference.
Conclusions
Lines 248-249 (Therefore, creating a TGS model for a single athletic event is recommended): It is worth mentioning that the data from the present study suggest this for strength athletes, but that larger and more homogeneous cohorts are necessary for this conclusion. In some cases it will be okay to put together more than one athletic event.
Author Response
The aim of this study was to explore the genetic score in Japanese athletes, particularly weightlifters. The proposal is interesting and well designed, but there are a number of points that need to be adjusted (listed below).
====
Thank you for your comment. We have substantial revised a paper according to reviewer’s comments.
==============
Abstract
Lines 14-15: It appears that the second and third proposals are the same (study/validate the derived models, WCM and WPM, in an independent cohort of strength/power athletes). If this is true, the two can be combined to facilitate interpretation.
=======
Thank you for your suggestion. We have revised the abstract.
(Page1, Line11 – Page1, Line22)
====
Introduction
Lines 32-34 (While a single polymorphism is responsible for muscle strength variation…): It is important to note that a single polymorphism is partially responsible for muscle strength variation.
=====
Thank you for your comment. We have changed sentence.
Ahmetov et al. [1, 2] reviewed the association between strength and power, athletic status, and gene markers, and reported strength-elated and power-related least 38 and 24 markers. While a single polymorphism is partially responsible for muscle strength variation, however, the phenotypic influence of these single gene polymorphisms may be little.
(Page 2, Line 42 – Page 2, Line 44)
=====
Materials and Methods
Lines 71-72: How many men and women are there in the control group? (similar to that described for athletes)
====
Thank you for your comment. We have added information in participants.
Overall, 192 Japanese weightlifters (113 men and 79 women) and 416 controls (151 men; height: 169.0 ± 6.3 cm, weight: 68.5 ± 9.6 kg, age: 51.0 ± 19.07 year, 265 women; height: 156.9 ± 8.1 cm, weight: 54.8 ± 8.1 kg, age: 53.0 ± 17.4 year) living in Tokyo and surrounding areas were recruited in this study 1.
(Page 3, Line 91 – Page 3, Line 101)
=====
Line 75: What is the competitive level of the athletes included in study 2? (international level or only national). It is worth noting that a general description of the study design may be interesting. For example, highlighting that the athletes from study 2 were used only to validate the score derived from study 1.
====
Thank you for your comment. We have added information in power-oriented athletes.
In study 2, 177 strength and power athletes, and 416 controls partici-pated. The strength and power athlete group comprised 101 wrestlers (87 men, 14 women; 55 international and 46 national level), 40 men powerlifters (16 international and 39 national level), and 36 national level throwers (17 men, 19 women).
(Page 3, Line 98 – Page 3, Line 101)
====
Lines 81-83: Were only polymorphisms with minor allele frequency > 5% in the Japanese population considered? It wasn't clear.
====
Thank you for your comment. The genetic polymorphisms in this study were minor alleles with more than 5% genetic polymorphisms. Since the sample size of weightlifters was not large, genetic polymorphisms with a certain frequency were employed.
=====
Lines 83-84 or 88 (Overall, 14 polymorphisms were found to be candidate genes associated with muscle strength and power or Therefore, we selected 12 polymorphisms): Here, it would be interesting to write the name of the genes + rs# of each variant used.
===
Thank you for your suggestion. We have added sentence in the methods at “Genotyping”. (Page 3, Line 102 – Page 3, Line 110)
===
Results
Lines 143-144 (…only three polymorphisms were adopted in this study): It is confusing to understand why it is saying that only 3 polymorphisms were adopted in the study, while the TGS was calculated with the 12 polymorphisms. What was actually considered or disregarded in the main analyzes based on the Hardy-Weinberg equilibrium could be better described.
=====
Thank you for your comment. We have revised the sentence, in fact, 12 polymorphisms were considered. (Page 5, Line 205 – Page 5, Line 206)
Figures: The axes title was difficult to read.
Thank you for your suggestion. We have revised the fig.
(Page8 fig.1 and fig.2, Page9, fig.3)
====
Line 154 (…no significant difference between weightlifters and controls (control: 39.18 ± 19.33, weightlifter: 40.62 ± 20.8) as shown in Fig. 1 E and F): The description is wrong because in Fig 1 F there are data with statistical difference.
=====
Thank you for your suggestion. We have major revised the figure.
(Page8 fig.1 and fig.2)
=====
Conclusions
Lines 248-249 (Therefore, creating a TGS model for a single athletic event is recommended): It is worth mentioning that the data from the present study suggest this for strength athletes, but that larger and more homogeneous cohorts are necessary for this conclusion. In some cases it will be okay to put together more than one athletic event.
====
Thank you for your suggestion. We have revised the conclusions.
Six genetic polymorphisms related to weightlifter status could be identified. The TGS model generated from those genetic polymorphisms was higher in elite weightlifters compared to controls. However,the TGS model using six polymorphisms could not be applied to weightlifting performance and status in other power athletes. (Page13, Line 419 – Page13, Line 422)
====
Reviewer 2 Report
In order to assess whether the study presents sufficient quality to be published, major improvements are needed:
Abstract and introduction are unclear. Authors must be improved the justification way and the concepts that they want to consider. Many sentences are confusing (for example: Lines 32-34, 39-42) and I do not understand the study number three.
Also reference 1 is not Ahmetov et al.
I think the WCM and WPM models are inconsistent, and it is difficult to understand why the authors no consider all the SNPs for these two models.
In reference to material and methods: In Line 71. Are you analyzed 416 or 417controls? Where and how were recruited the control and the athletes? More information is necessary. In addition, the committee reference number is missing.
NOS3 polymorphism analysis should be performed
Descriptive information about the participants is missing. Genotype score is poorly presented. Again, it is difficult to understand the justification the Genotype Score Models and the reason for separated in two different studies the populations.
I think this study is simple: the authors have controls, strength and power athletes and weightlifters and analyzed in these groups 12 SNPs to calculate a Genotype Score. It is better to explain the study in this way that do it mixed and unclear.
Once, the results must be improved.
I think this study is simple: the authors have controls, strength and power athletes and weightlifters and analyzed in these groups 12 SNPs to calculate a Genotype Score. It is better to explain the study in this way that do it mixed and unclear.
Once, the results must be improved.
Author Response
In order to assess whether the study presents sufficient quality to be published, major improvements are needed:
====
Thank you for your comment. We changed substantially in this paper according to your comments.
====
Abstract and introduction are unclear. Authors must be improved the justification way and the concepts that they want to consider. Many sentences are confusing (for example: Lines 32-34, 39-42) and I do not understand the study number three.
====
We have revised the sentence.
This study aimed to investigate the relationship between power-oriented genetic polymor-phisms and weightlifting status, and create a total genotype score (TGS), and validate the asso-ciation between TGS models and power-oriented athletes. First, 192 weightlifters and 416 controls were studied, and 12 polymorphisms that have previously been associated with strength and power status and phenotype were genotyped using the TaqMan SNP genotyping assay. We calculated the TGS for the 12 (PWM) and 6 polymorphisms (WRM) based on the case-control study. Second, the TGS of WRM was compared for 177 strength and power athletes and 416 controls. There was no significant difference in PWM score between weightlifters and controls. Whole weightlifters and elite weightlifters had higher WRM scores than those of the controls. There was no significant difference in WRM between the power-oriented athletes and controls. Our study was able to create a TGS model for weightlifters based on case control. However, the TGS model could not be applied to other power-oriented athletes.
(Page1, Line 11 – Page 1, Line 22)
=====
Also reference 1 is not Ahmetov et al.
====
We have revised the sentence.
Ahmetov et al. [1] reviewed the association between strength and power, athletic status, and gene markers and reported at least 38 and 24 strength-related and power-related markers. While a single polymorphism is partially responsible for muscle strength variation, the phenotypic influence of these single gene polymorphisms may be limited.
(Page 2, Line 43 – Page 2, Line 45)
====
I think the WCM and WPM models are inconsistent, and it is difficult to understand why the authors no consider all the SNPs for these two models.
=====
Thank you for your comment. We have revised purpose and results.
First, we conducted a case-control study of 12 polymorphisms, which are associated with strength and power and weightlifters. The results showed that six genetic polymorphisms were associated with weightlifters. Secondly, six gene polymorphisms were scored and compared between weightlifters and controls. Thirdly, we compared case controls and scores in power athletes with genetic polymorphisms associated with weightlifters.
Purpose: (Page 2, Line 82 – Page 2, Line 86)
====
In reference to material and methods: In Line 71. Are you analyzed 416 or 417controls? Where and how were recruited the control and the athletes? More information is necessary. In addition, the committee reference number is missing.
======
Thank you for your comment. We have added information in participants and committee reference number in Institutional Review Board Statement.
Overall, 192 Japanese weightlifters (113 men and 79 women) and 416 controls (151 men; height: 169.0 ± 6.3 cm, weight: 68.5 ± 9.6 kg, age: 51.0 ± 19.07 year, 265 women; height: 156.9 ± 8.1 cm, weight: 54.8 ± 8.1 kg, age: 53.0 ± 17.4 year) living in Tokyo and surrounding areas were recruited in this study 1.
(Page 3, Line 91 – Page 3, Line94)
Institutional Review Board Statement: The study was conducted according to the guidelines of the Declaration of Helsinki and approved by the Ethics Committee of the Federal Research and Clinical Center of Physical-Chemical Medicine and by the ethics committees of the Nippon Sport Science University(020-G03)
(Page 13, Line 438 – Page 13, Line 441)
=====
NOS3 polymorphism analysis should be performed
====
Thank you for your comment. In fact, we conducted several times analysis of NOS3 rs2040744 polymorphism. However, it was deviated from Hardy-Weinberg equilibrium in the control and whole participants and difference from common frequency in database, therefore, NOS3 rs2040744 was excluded. The NOS3 rs2040744 polymorphism was determined to be inaccurate for PCR analysis. However, we excluded this sentence.
=====
Descriptive information about the participants is missing. Genotype score is poorly presented. Again, it is difficult to understand the justification the Genotype Score Models and the reason for separated in two different studies the populations.
=====
Thank you for your suggestion. We have added information of participants and genotype score. One of the purposes of the study is to determine whether genetic profiles differ among events that are classified in the same strength and power athletes. We have clearly written this purpose.
(Page 3, Line 91 – Page 3, Line 101) and (Page 4, Line 150 – Page 5, Line 176)
====
I think this study is simple: the authors have controls, strength and power athletes and weightlifters and analyzed in these groups 12 SNPs to calculate a Genotype Score. It is better to explain the study in this way that do it mixed and unclear.
====
Thank you for your comment. We revised whole paper according to your comments.
====
Once, the results must be improved.